# Cilia-mediated Hedgehog signaling controls form and function in the mammalian larynx

Jacqueline M Tabler[1†], Maggie M Rigney[1†], Gordon J Berman[2], Swetha Gopalakrishnan[3], Eglantine Heude[3], Hadeel Adel Al-lami[4], Basil Z Yannakoudakis[4], Rebecca D Fitch[1], Christopher Carter[1], Steven Vokes[1], Karen J Liu[4], Shahragim Tajbakhsh[3], SE Roian Egnor[5], John B Wallingford[1*]

[1]Department of Molecular Biosciences, University of Texas at Austin, Austin, United States; [2]Department of Biology, Emory University, Atlanta, United States; [3]Stem Cells and Development, CNRS UMR3738, Department of Developmental and Stem Cell Biology, Institut Pasteur, Paris, France; [4]Department of Craniofacial Development and Stem Cell Biology, King's College London, London, United Kingdom; [5]Janelia Research Campus, Howard Hughes Medical Institute, Ashburn, United states

*For correspondence:
wallingford@austin.utexas.edu

[†]These authors contributed equally to this work

Competing interests: The authors declare that no competing interests exist.

**Abstract** Acoustic communication is fundamental to social interactions among animals, including humans. In fact, deficits in voice impair the quality of life for a large and diverse population of patients. Understanding the molecular genetic mechanisms of development and function in the vocal apparatus is thus an important challenge with relevance both to the basic biology of animal communication and to biomedicine. However, surprisingly little is known about the developmental biology of the mammalian larynx. Here, we used genetic fate mapping to chart the embryological origins of the tissues in the mouse larynx, and we describe the developmental etiology of laryngeal defects in mice with disruptions in cilia-mediated Hedgehog signaling. In addition, we show that mild laryngeal defects correlate with changes in the acoustic structure of vocalizations. Together, these data provide key new insights into the molecular genetics of form and function in the mammalian vocal apparatus.

## Introduction

Vocal communication is fundamental to social interaction. Indeed, the voice is so crucial to our quality of life that the neurobiology of speech and language has been hotly studied for decades, as has the developmental biology of the ear. These bodies of work stand in surprising contrast to our still rudimentary understanding of the developmental biology of the organs of vocalization, the larynx and vocal folds. This is true despite the fact that most animal vocalizations, including human speech, are critically dependent upon the careful control of airflow though the larynx. In fact, larynx and vocal fold morphology and elasticity are key factors influencing vocalization even in animals with widely divergent mechanisms of sound production (e.g. audible vocalizations in humans, ultrasound in rodents).

This deficit in our understanding of laryngeal and vocal fold development is significant, because many people who are capable of normal *speech* still cannot communicate due to defects in *voice* (e.g. problems with pitch, loudness, etc.). Some voice defects arise from acute insults, such as insufficient hydration of the vocal folds in *laryngitis sicca* or vocal fold hemorrhages resulting from blood vessel ruptures (*Aronson and Bless, 2009*). Other conditions are hereditary and chronic, such as

**eLife digest** Nearly all animals communicate using sound. In many cases these sounds are in the form of a voice, which in mammals is generated by a specialized organ in the throat called the larynx. Millions of people throughout the world have voice defects that make it difficult for them to communicate. Such defects are distinct from speech defects such as stuttering, and instead result from an inability to control the pitch or volume of the voice. This has a huge impact because our voice is so central to our quality of life.

A wide range of human birth defects that are caused by genetic mutations are known to result in voice problems. These include disorders in which the Hedgehog signaling pathway, which allows cells to exchange information, is defective. Projections called cilia that are found on the outside of many cells transmit Hedgehog signals, and birth defects that affect the cilia (called ciliopathies) also often result in voice problems.

Although the shape of the larynx has a crucial effect on voice, relatively little is known about how it develops in embryos. Mice are often studied to investigate how human embryos develop. By studying mouse embryos that had genetic mutations similar to those seen in humans with ciliopathies, Tabler, Rigney et al. now show that many different tissues interact in complex ways to form the larynx. A specific group of cells known as the neural crest was particularly important. The neural crest helps to form the face and skull and an excess of these cells causes face and skull defects in individuals with ciliopathies. Tabler, Rigney et al. show that having too many neural crest cells can also contribute towards defects in the larynx of mice with ciliopathies, despite the larynx being in the neck.

Further investigation showed that the Hedgehog signaling pathway was required for the larynx to develop properly. Furthermore, recordings of the vocalizations of the mutant mice showed that they had defective voices, thus linking the defects in the shape of the larynx with changes in the vocalizations that the mice made.

Overall, Tabler, Rigney et al. show that mice can be used to investigate how the genes that control the shape of the larynx affect the voice. The next step will be to use mice to investigate other genetic defects that cause voice defects in humans. Further research in other animals could also help us to understand how the larynx has evolved.

those arising from mutations in genes encoding the extracellular matrix protein Elastin (*Vaux et al., 2003*; *Watts et al., 2008*). All of these conditions impact the voice, thereby impacting patients' well-being.

A wide array of human birth defect syndromes also involve voice defects, and prominent among these are disorders stemming from failure of the Hedgehog (HH) signaling pathway, an evolutionarily conserved mechanism for cell-cell communication (*Briscoe and Thérond, 2013*). For example, Pallister-Hall Syndrome is caused by mutations in *Gli3*, a key transducer of HH signals. These patients have hoarse and/or growling voices, and they frequently exhibit laryngeal clefts and bifid epiglottis (*Hall et al., 1980*; *Tyler, 1985*). Pallister-Hall Syndrome is known for its variable expressivity, and accordingly, this disorder is also associated with milder laryngeal anomalies (*Ondrey et al., 2000*). Importantly, laryngeal and voice defects are not limited to *Gli3* mutation, but have also been associated with mutation in the related factor *Gli2* (*França et al., 2010*), in the Shh transducer *Kif7* (*Putoux et al., 2012*; *Walsh et al., 2013*), and in *Shh* itself (*Cohen, 2004*).

Cilia are essential organelles for transduction of HH signals (*Goetz and Anderson, 2010*), and as a result, voice defects are also commonly associated with ciliopathies, human diseases that share an etiology of defective cilia structure or function (*Hildebrandt et al., 2011*). For example, a breathy, high-pitched voice is a diagnostic criterion for Bardet-Biedl and Oral-Facial-Digital syndromes, while hoarse voices are diagnostic for Joubert Syndrome (*Beales et al., 1999*; *Garstecki et al., 1972*; *Hayes et al., 2008*; *Maria et al., 1999*; *Rimoin and Edgerton, 1967*). Laryngeal defects such as laryngeal stenosis and bifid epiglottis are also common features of other ciliopathies (*Carron, 2006*; *Hayes et al., 2008*; *Silengo et al., 1987*; *Steichen-Gersdorf et al., 1994*; *Stevens and Ledbetter, 2005*).

Understanding the molecular genetic basis for voice disorders in human birth defect patients is not the only factor motivating a deeper study of laryngeal developmental biology. Indeed, vocal communication is ubiquitous in tetrapod animals, impacting a wide array of behaviors. For example, the Panamanian Tungara frog creates a complex, multi-tonal call that critically influences female mate choice, and this call requires a sexually dimporphic elaboration of the male larynx, the developmental basis of which is entirely unknown (*Griddi-Papp et al., 2006*; *Ryan and Drewes, 1990*). So too is the morphology of the songbird syrinx central to sound production, yet almost nothing is known of the developmental biology of this functional cognate of the larynx, despite the key role of bird song as a model for the study of acoustic communication. Likewise, the larynx of mice is central to their production of ultrasonic vocalizations throughout life. Despite the widespread use of mice for studies of developmental biology, the molecular genetics of mouse laryngeal development remain only cursorily poorly defined (e.g. [*Böse et al., 2002*; *Lungova et al., 2015*]). Clearly, a deeper understanding of the molecular genetic basis of laryngeal patterning and morphogenesis will inform our understanding of vertebrate animal behaviors involving acoustic communication.

In mammals, the larynx and vocal folds are comprised of an elaborate mixture of cartilages, muscles, nerves, and connective tissue (*Harrison, 1995*; *Henick, 1993*; *Lungova et al., 2015*). The flanged circle of the cricoid cartilage, along with the C-shaped thyroid cartilage and intervening paired arytenoid cartilages provide the core of the laryngeal skeleton (*Figure 1*, blue, yellow, purple). Anchored to these are the vocal folds, which are in turn comprised of paired *cricoarytenoid*, *thyroarytenoid, cricothyroid* and *vocalis* muscles (*Figure 1*, pink, magenta, grey), as well as paired vocal ligaments (*Figure 1*, dark blue) and associated loose mesenchyme which we designate as the thyroglottal connective tissue (*Figure 1*, green). The general laryngeal structure is similar across the mammals (*Harrison, 1995*; *Kaufman, 1992*; *Roberts, 1975a*; *Thomas et al., 2009*), though rodents communicate most commonly in the ultrasonic range, using a mechanism for sound production that is distinct from that generating audible sound (*Mahrt et al., 2016*; *Roberts, 1975b*). Importantly however, diverse aspects of rodent ultrasound production parallel those of audible vocalizations in other mammals, including tight control of laryngeal muscle activity and mechanical properties of the vocal folds (*Riede, 2011*, *2013*).

Here, we have characterized severe defects in laryngeal and vocal fold development in ciliopathic *Fuz* mutant mice, as well as similar though less severe defects in *Gli3* mutant mice. To understand the developmental trajectory of these defects, we first performed directed genetic fate mapping of the mouse larynx. We defined the embryonic origins for diverse tissues in the larynx, and also show that that laryngeal defects in both *Fuz* and *Gli3* mutants stem from an invasion of excess neural crest. Finally, we show that viable heterozygous *Gli3* mutant mice display quantitative changes in the morphology of the vocal apparatus accompanied by significant changes in the acoustic structure of their ultrasonic vocalizations. Together, these findings provide an improved foundation for molecular genetic studies of development in the mammalian vocal apparatus, establish a genetic animal model for understanding human congenital laryngeal and voice defects, and demonstrate that excess neural crest is a common etiology underlying diverse Hedgehog-related craniofacial defects.

## Results

### Laryngeal and vocal fold defects in mouse models of human ciliopathies and Pallister-Hall syndrome

Voice and laryngeal defects are common in ciliopathies, including Oral-facial-digital Syndrome Type 6 (*Hayes et al., 2008*). Recently, we showed that mutation of genes encoding the Ciliogenesis and Planar Cell Polarity effector (CPLANE) proteins results in OFD phenotypes in mice (*Tabler et al., 2013*; *Toriyama et al., 2016*). We therefore examined the larynx of mice lacking the CPLANE component *Fuz* as a first step towards understanding the developmental basis for ciliopathic larynx and voice defects. We observed severe malformation of the laryngeal cartilages in *Fuz* mice, as well as severely disorganized and hypoplastic vocal fold musculature (*Figure 2A,B*, *Figure 2—figure supplement 1B–C*). No glottic space could be identified in the mutants, and the entire larynx was instead filled with an accumulation of loose connective tissue (*Figure 2B,B'*, *Figure 2—figure supplement 1C–C*). This severe derangement of the larynx in *Fuz* mutants prevented identification of specific laryngeal cartilages, making interpretation of these sections challenging. However, in frontal

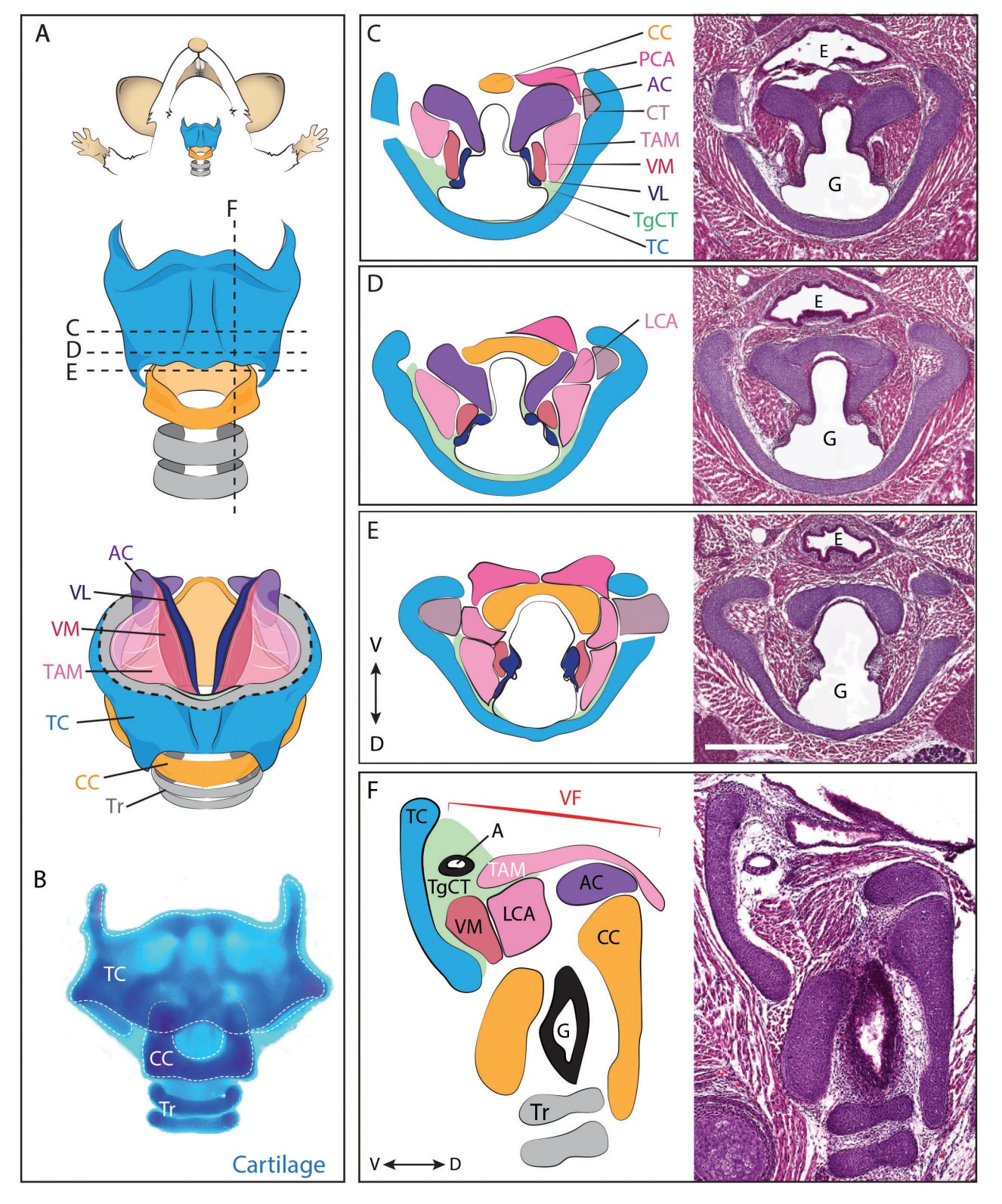

**Figure 1.** Anatomy of the mouse larynx. (**A**) Diagram representing ventral view of mouse laryngeal anatomy. Dashed lines indicate sectional plane represented in panels **C–F**. (**B**) Ventral view of an excised adult larynx stained with alcian blue marking cartilage. (**C–E**) H&E staining of horizontal sections of E18.5 mouse larynx. Sectional plane is indicated in **A**. Diagrams indicate anatomy observed in sections. (**F**) H&E staining of sagittal section of E18.5 mouse larynx. Diagram indicates anatomy represented in section. Scale bar indicates 500 μm. Abbreviations: (AC) Arytenoid Cartilage, (CC)
*Figure 1 continued on next page*

*Figure 1 continued*

Cricoid Cartilage, (CT) Cricothyroid muscle, (E) Esophagus, (G) Glottis, (L) Larynx, (LCA) Lateral Cricoarytenoid muscle, (PCA) Posterior Cricoarytenoid muscle, (T) Tongue, (TAM) Thyroarytenoid Muscle, (TC) Thryoid Cartilage, (TgCT) Thyroglottal connective tissue, (Tr) Trachea, (VL) Vocal Ligament, (VM) Vocalis Muscle, (VF) Vocal fold. (V) and (D) indicate dorso-ventral axes.

sections of control mice, we can identify four distinct cartilage elements (*Figure 2—figure supplement 1A,B,B'*), including the three laryngeal cartilages and the hyoid cartilage (*Kaufman, 1992*), while by contrast, we observe only what appears to be a single severely disordered cartilage element in similar frontal sections of *Fuz* mutants (*Figure 2—figure supplement 1C,C'*).

Cilia-mediated Hedgehog signals influence the processing of both the Gli2 and Gli3 transcription factors (*Haycraft et al., 2005*), so we reasoned that mutation of either one of those two factors may generate milder, more interpretable laryngeal phenotypes. *Gli3* mutant mice provide useful models for Gli-related human birth defects (*Böse et al., 2002*; *Hui and Joyner, 1993*), so we examined the *Gli3^{xt-J}* mice. *Gli3* homozygous mutant mice developed with overt laryngeal defects, and as predicted, these were far milder than those in *Fuz* mice (*Figure 2B,C*). Unlike *Fuz* mutants, the laryngeal cartilages appeared normal in *Gli3* mutants and the glottis was evident. However, *Gli3^{-/-}* mice consistently developed with an aberrant accumulation of Thyroglottal Connective Tissue (TgCT) around

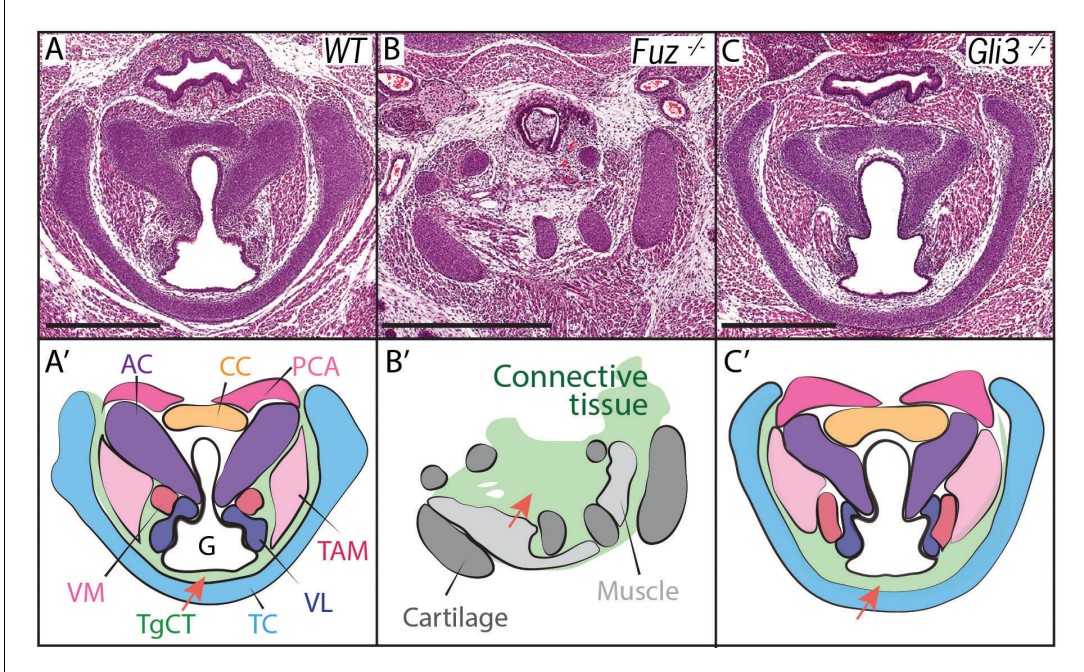

**Figure 2.** Laryngeal anatomy is disrupted in *Fuz* and *Gli3* mutants. (A–C) H and E staining of horizontal sections of E18.5 larynges. (A'–C') Diagrams of anatomy shown in (A–C). *Fuz* mutant larynges (B–B') are significantly altered compared to controls, (A–A'). Connective tissue in mutants (light green, (C') is increased in mutants compared to controls (A–A'), while cartilage and muscle are irregularly patterned. *Gli3^{-/-}* larynges are less altered than *Fuz^{-/-}* embryos compared to controls (A–A'). Thyroglottal Connective tissue appears increased in *Gli3* mutants (light green, red arrow, (C'). Scale bars indicate 500 μm. Abbreviations: (AC) Arytenoid Cartilage, (CC) Cricoid Cartilage, (CT) Cricothyroid muscle, (G) Glottis, (L) Larynx, (LCA) Lateral Cricoarytenoid muscle, (PCA) Posterior Cricoarytenoid muscle, (T) Tongue, (TAM) Thyroarytenoid Muscle, (TC) Thryoid Cartilage, (TgCT) Thyroglottal connective tissue, (Tr) Trachea, (VL) Vocal Ligament, (VM) Vocalis Muscle, (VF) Vocal fold. (V) and (D) indicate dorso-ventral axes.

The following figure supplements are available for figure 2:

**Figure supplement 1.** Fgf8 reduction in Fuz mutants partially rescues laryngeal phenotypes.

**Figure supplement 2.** Wnt1Cre driven deletion of Fuz does not affect laryngeal morphology.

the vocal folds and particularly between the ventral limit of the glottis and the thyroid cartilage (*Figure 2C,C'*, arrow), which was only 2–4 cells wide in normal mice, but was substantially expanded in *Gli3$^{-/-}$* mice. In addition, we observed a decrease in the ventral extension of at least the *thyroarytenoid* muscles with a concomitant expansion of loose mesenchyme between these muscles and the thyroid cartilage (*Figure 2C,C'*). Interestingly, the accumulated mesenchyme in *Gli3* mutants appeared histologically similar to that seen in the more severely deranged larynx of *Fuz* mutants. Together, these data suggest a potential role for cilia-mediated Gli signaling in the patterning of the mammalian larynx.

## Genetic fate mapping of the mouse larynx and vocal folds

We next sought to understand the developmental trajectory of laryngeal defects in our mouse models, but this goal was hampered by the paucity of fate mapping data for the larynx. Indeed, there have as yet been only tangential reports of the developmental origins of tissues in the larynx, and even these results are not entirely consistent. For example, one study reports that the major laryngeal cartilages are of a neural crest origin (*Matsuoka et al., 2005*), but that mapping is surprising in light of other mouse genetic studies that suggest a mixed lineage (e.g. *Jeong et al., 2004*; *Mori-Akiyama et al., 2003*). Moreover, at least some laryngeal cartilages have a mesodermal origin in birds (*Evans and Noden, 2006*; *Noden, 1986a*). Recent studies using clonal or lineage analysis in mice suggest a relationship between some laryngeal muscles and the branchiomeric neck muscles (*Gopalakrishnan et al., 2015*; *Lescroart et al., 2015*), but information is lacking on the origin of the muscles and ligaments that comprise the vocal folds themselves.

We first used genetic fate mapping with *Wnt1$^{Cre}$:R26$^{mT/mG}$* to map the descendants of neural crest cells in the larynx (*Chai et al., 2000*). Histological sections revealed distinct lineages for the three laryngeal cartilages. While the thyroid cartilage was prominently labeled by *Wnt1$^{Cre}$::mGFP*, the arytenoid and cricoid cartilages were unlabeled, suggesting they are not crest-derived (*Figure 3B*). Strikingly, we observed a mixed lineage even within the single thyroid cartilage element; the medial caudal-most portion of the thyroid cartilage was consistently unlabeled by *Wnt1$^{Cre}$::mGFP* (*Figure 3D,D', H,H'*). This result was unexpected, so we confirmed it using an alternative promoter to label neural crest and an alternative reporter allele (*Li et al., 2000*). *Pax3$^{Cre}$:R26$^{Tomato}$* mice also displayed strong label throughout most of the thyroid cartilage, but not in the caudal ventral midline; label was also absent from the cricoid and arytenoid cartilages (*Figure 3F–F'*).

We then performed a similar analysis using *Mesp1$^{Cre}$;R26$^{mT/mG}$* and *Mesp1$^{Cre}$;R26$^{Tomato}$* to map mesoderm-derived tissues (*Saga et al., 1999*) (*Figure 4*). *Mesp1$^{Cre}$* clearly labeled the ventral midline of the caudal thyroid cartilage, the region that was unlabeled by *Wnt1$^{Cr}$; R26$^{mT/mG}$* (*Figure 4C, D*), arguing that this single cartilage arises from a mixture of neural crest and mesoderm. *Mesp1$^{Cre}$; R26$^{mT/mG}$* and *Mesp1$^{Cre}$;R26$^{Tomato}$* lineage analysis also revealed a mesodermal origin for the cricoid cartilage and arytenoid cartilages (*Figure 4A,C*).

In the vocal folds themselves, *Wnt1$^{Cre}$* drove GFP expression in the vocal ligaments, which are key elements for vocal fold apposition during sound production (*Figure 3A,B,B'*). Not surprisingly, the muscles of the vocal fold (*vocalis*, *thyroarytenoid)* were not labeled by *Wnt1$^{Cre}$::mGFP*, but were robustly labeled by *Mesp1$^{Cre}$* lineage (*Figure 3A,B*; *Figure 4A,D*). We also observed threads of *Wnt1$^{Cre}$* labeled, crest-derived cells interspersed within the vocal fold muscles (*Figure 3A,B,B'*). Some of these cells were identified as neurons by acetylated tubulin immunostaining (not shown), consistent with reports of vocal fold paralysis in neurocristopathies such as CHARGE syndrome (*Jongmans et al., 2006*; *Siebert et al., 1985*). Other crest-derived cells in the vocal fold likely represent the fascia separating the vocal muscles. Finally, *Wnt1$^{Cre}$;R26$^{mT/mG}$* labeled the TgCT, the thin layer of connective tissue separating the ventral aspect of the glottis from the thyroid cartilage (*Figure 3A,B*).

Finally, because cranial and axial muscles develop *via* distinct genetic programs and originate from different mesodermal populations (*Sambasivan et al., 2011*), we sought to determine which of the mesoderm lineages contributes to the muscles of the vocal folds. In cranial mesoderm, *Isl1*-positive myogenic progenitors contribute to the formation of head muscles (*Harel et al., 2009*; *Nathan et al., 2008*), while *Pax3*-positive cells in the somitic mesoderm give rise to trunk and limb musculature. Analysis of the *Pax3;R26$^{Tomato}$* lineage suggested that the muscles of the vocal folds, marked by Desmin immunostaining, were not *Pax3*-derived and, thus, not of somitic origin

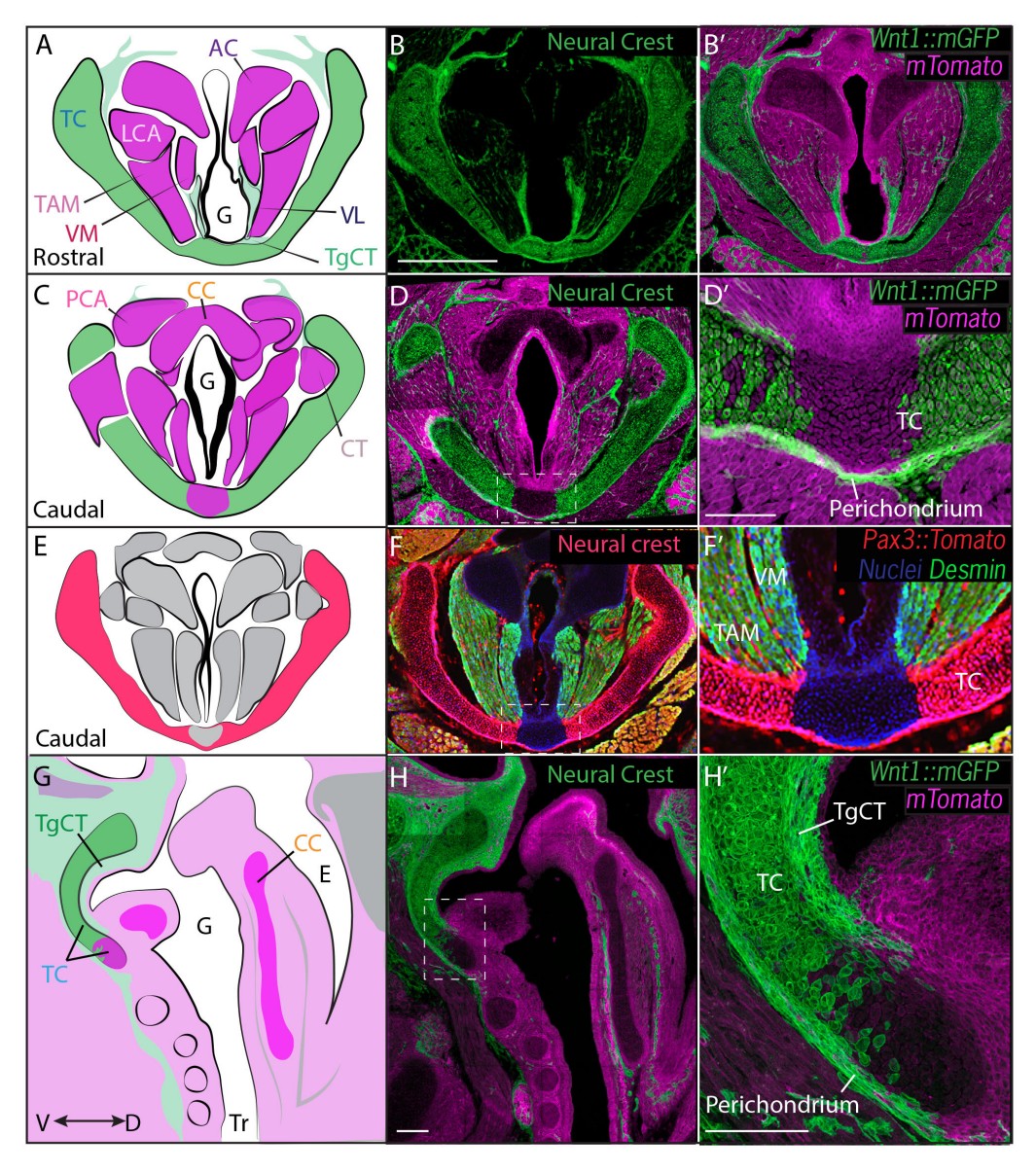

**Figure 3.** Thyroid cartilage and vocal ligament are mostly neural crest derived. (**A**) Diagram representing anatomy in (**B–B'**). (**B–B'**) Horizontal section of rostral E18.5 *Wnt1^(Cre/+)*; *R26^(mTmG)*larynx. Neural crest derivatives are labeled in green while other tissues are labeled with Magenta. (**B–C**) Scale bar indicates 200 μm. (**C**) Diagram representing anatomy in (**D–D'**) Horizontal section of caudal E18.5 *Wnt1^(Cre/+)*; *R26^(mTmG)*larynx. (**E**) Diagram representing anatomy in (**F–F'**). (**F–F'**) Horizontal caudal section of E18.5 *Pax3^(Cre/+)*; *R26^(tomato)* larynx that is also immunostained for Desmin which marks muscle (Green and yellow in cells that have also expressed *Pax3*) and nuclei (Blue). (**G**) Diagram representing anatomy in (**H–H'**). (**H–H'**) Sagittal section of E18.5 *Wnt1^(Cre/+)*; *R26^(mTmG)*larynx. Thyroglottal Insets are indicated with white dotted box. (**F–F'**) Scale bars indicate 100 μm. Abbreviations: (AC) Arytenoid Cartilage, (CC) Cricoid Cartilage, (CT) Cricothyroid muscle, (E) Esophagus, (G) Glottis, (L) Larynx, (LCA) Lateral Cricoarytenoid muscle, (PCA) Posterior Cricoarytenoid muscle, (TAM) Thyroarytenoid Muscle, (TC) Thryoid Cartilage, (TgCT) Thyroglottal connective tissue, (Tr) Trachea, (VL) Vocal Ligament, (VM) Vocalis Muscle, (VF) Vocal fold. (V) and (D) indicate dorso-ventral axes.

(*Figure 3F*). In contrast, *Isl1^(Cre)*;*R26^(mT/mG)* mice showed that all muscles of the vocal folds labeled by the Desmin immunostaining (*Figure 4A*) were derived from the *Islet1; R26^(mT/mG)* lineage (*Figure 4B*), demonstrating their cranial mesoderm origin.

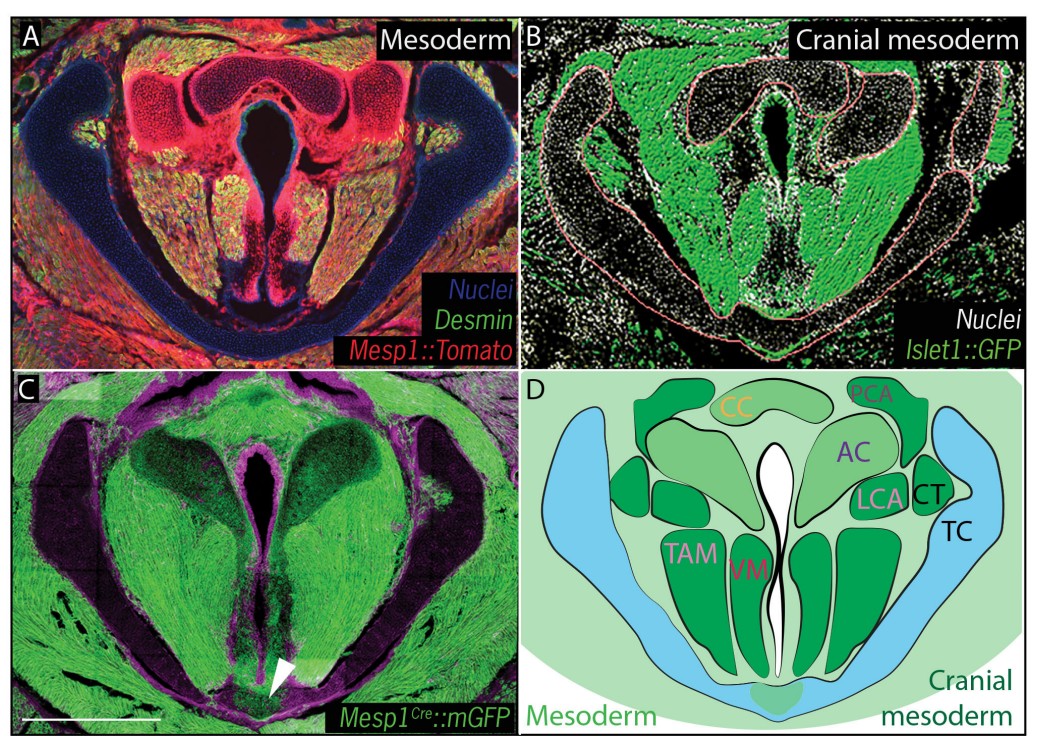

**Figure 4.** Vocal fold muscles are from cranial mesodermal origin. (**A**) Horizontal section of rostral E18.5 *Mesp1^Cre^; R26^Tomato^* larynx showing that the arytenoid and cricoid cartilages, and Desmin-positive vocal fold muscles are derived from mesoderm. (**B**) Horizontal section of rostral E18.5 *Islet1^Cre^; R26^mTmG^* larynx indicating that all the vocal fold muscles are of cranial mesoderm origin. (**C**) Horizontal section of E18.5 larynx of *Mesp1^Cre^; R26^mTmG^* mouse showing the ventral part of the thyroid cartilage derived from mesoderm. (**D**) Diagram of anatomy represented in (**A–C**). The mesoderm derivatives are labeled in light green while the specific muscular cranial mesoderm derivatives are labeled in dark green. Scale bars indicate 100 µm. Abbreviations: (AC) Arytenoid Cartilage, (CC) Cricoid Cartilage, (CT) Cricothyroid muscle, (LCA) Lateral Cricoarytenoid muscle, (PCA) Posterior Cricoarytenoid muscle, (TAM) Thyroarytenoid Muscle, (TC) Thryoid Cartilage, (Tr) Trachea,, (VM) Vocalis Muscle, (VF) Vocal fold.

Together, these data complement previous lineage analyses of the larynx in other species such as birds (*Evans and Noden, 2006*; *Noden, 1986a*) and provide the first comprehensive description of the developmental origins of tissues in mammalian larynx.

## Excess neural crest underlies defective vocal fold morphogenesis in Gli3 and fuz mutant mice

Our fate mapping of the normal mouse larynx provided us with a platform from which to explore the developmental basis for laryngeal defects in our mutant mice. To this end, we performed *Wnt1^Cre^* lineage labeling on *Fuz^-/-^* and *Gli3^-/-^* mice, focusing on cell lineages during initial morphogenesis of the larynx. Between E11.5 and E14.5, laryngeal morphogenesis proceeds in a surprisingly convoluted manner, with the previously patent lumen of the developing trachea becoming occluded by the formation of a structure known as the epithelial lamina. This epithelial lamina and tissues surrounding it then remodel into the vocal folds, and a new lumen forms that will ultimately constitute the glottis (*Henick, 1993*; *Lungova et al., 2015*; *Sañudo and Domenech-Mateu, 1990*). At E14.25, when the re-canalized glottis is already apparent in control mice, we found that condensing *Wnt1^Cre^* labeled neural crest cells were present at the site of the future thyroid cartilage and also in dorsally projecting streams presaging the neural crest-derived tissues in the vocal folds, such as ligaments, fascia and neurons, while other structures such as precursors of arytenoid and cricoid cartilage and future vocal fold muscles were unlabeled (*Figure 5A*).

At these stages, the entire region of the forming larynx in *Fuz^-/-^* mice was filled with *Wnt1^Cre^; R26^mT/mG^* positive, neural crest-derived cells; only scattered, individual unlabeled cells were

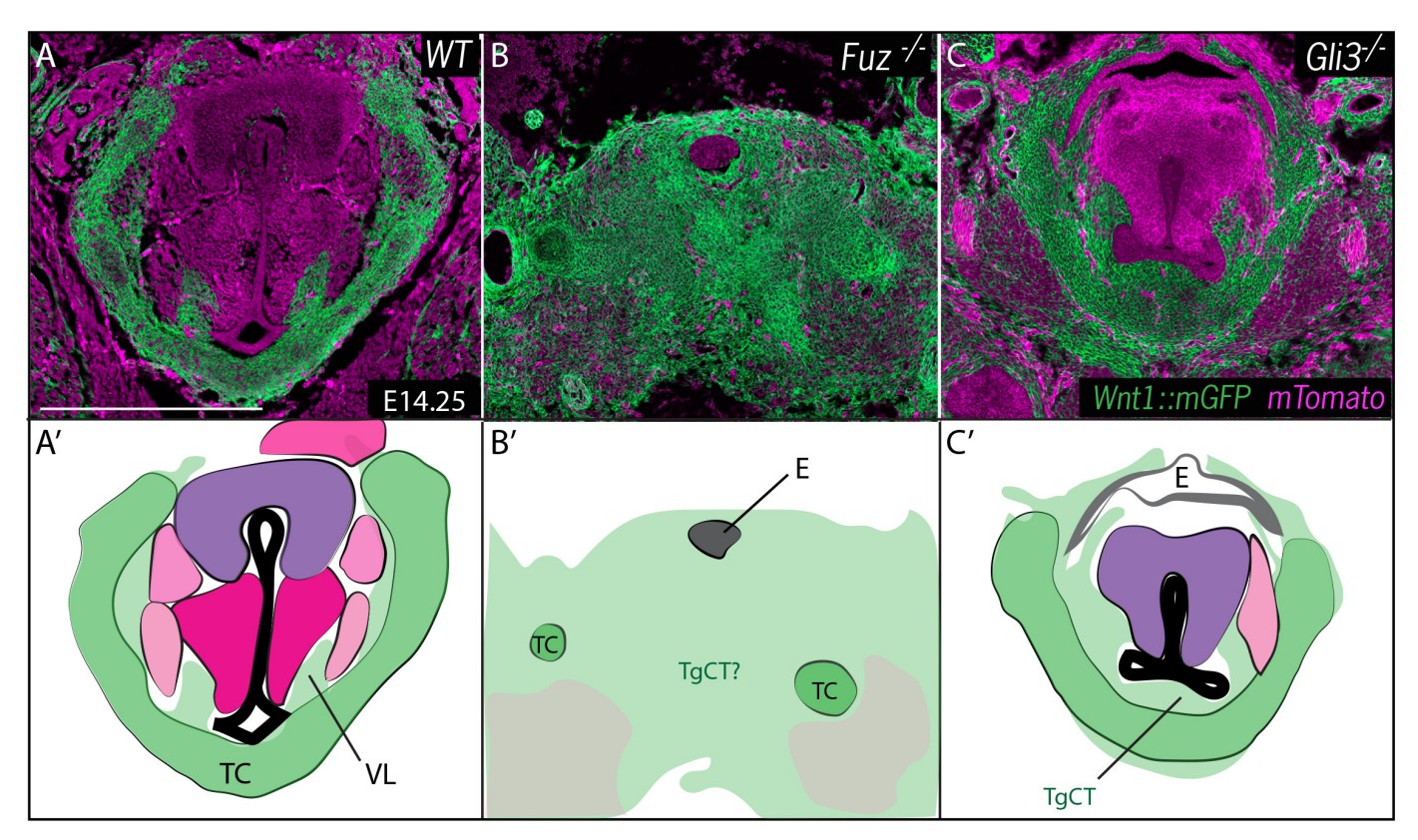

**Figure 5.** Neural crest is expanded in *Fuz* and *Gli3* mutant larynges. (**A–C**) Horizontal section of E14.25 larynges. (**A**) Wild Type *Wnt1*<sup>Cre</sup>*::mGFP* labeled larynx (**B**) *Fuz*<sup>-/-</sup>; *Wnt1*<sup>Cre/+</sup>; *R26*<sup>mTmG</sup> larynx. (**C**) *Gli3*<sup>-/-</sup>;*Wnt1*<sup>Cre/+</sup>; *R26*<sup>mTmG</sup>. Neural crest is labeled in green and other tissues in magenta. (**A'–C'**) Diagrams representing anatomy found in (**A–C**). Abbreviations: (AC) Arytenoid Cartilage, (CC) Cricoid Cartilage, (CT) Cricothyroid, (E) Esophagus, (G) Glottis, (L) Larynx, (LCA) Lateral Cricoarytenoid, (PCA) Posterior Cricoarytenoid muscle, (TAM) Thyroarytenoid Muscle, (TC) Thryoid Cartilage, (T) Tongue, (Tr) Trachea, (VL) Vocal Ligament, (VM) Vocalis Muscle, (VF) Vocal fold. (V) and (D) indicate dorsal ventral axes. Scale bars indicates 200 μm.

observed (*Figure 5B*), consistent with the very severe phenotype observed at later stages by H&E staining (*Figure 2B*). *Gli3*<sup>-/-</sup> mice displayed less severe phenotypes, with a substantial reduction of the glottic space accompanied by a striking excess of *Wnt1*<sup>Cre</sup>; *R26*<sup>mT/mG</sup> positive cells ventral to the glottis, in the area of the thyroid cartilage and TgCT (*Figure 5C*).

*Fuz* mutant mice rarely survive to E17.5, but at this stage in *Gli3*<sup>+/-</sup> and *Gli3*<sup>-/-</sup> mice, *Wnt1*<sup>Cre</sup> lineage mapping revealed a continued excess of neural crest-derived cells that were concentrated around the periphery of the glottis compared to controls (*Figure 6*). The accumulation of *Wnt1*<sup>Cre</sup>; *R26*<sup>mT/mG</sup> positive cells was especially pronounced in the TgCT separating the ventral edge of the glottis from the thyroid cartilage in *Gli3*<sup>-/-</sup> embryos (*Figure 6A,C*). An excess of neural crest-derived cells was also observed to disrupt the normal close association of the vocal muscles with the thyroid cartilage (*Figure 6C,C'*). These excess neural crest cells were found precisely in the position occupied by the aberrant mesenchyme observed by H and E staining in *Gli3*<sup>-/-</sup> mutants (*Figure 2C*; *Figure 6D', F'*), indicating that the cells are neural crest-derived.

These findings suggest that expanded neural crest disrupts laryngeal morphogenesis, which is of interested because we previously showed that an expansion of neural crest underlies palate defects in *Fuz* mutant mice and skull defects in both *Fuz* and *Gli3* mutant mice (*Tabler et al., 2013*, *2016*). In those instances, the defects can be rescued by genetic reduction of *Fgf8* gene dosage (*Tabler et al., 2013, 2016*). To ask if a similar mechanism acts in the larynx, we reduced the genetic dosage of *Fgf8* in *Fuz* mutants using the *Fgf8*<sup>LacZ</sup> knockin allele (*Ilagan et al., 2006*). Analysis of frontal sections revealed a partial rescue of *Fuz* mutant phenotype when *Fgf8* gene dosage is reduced;

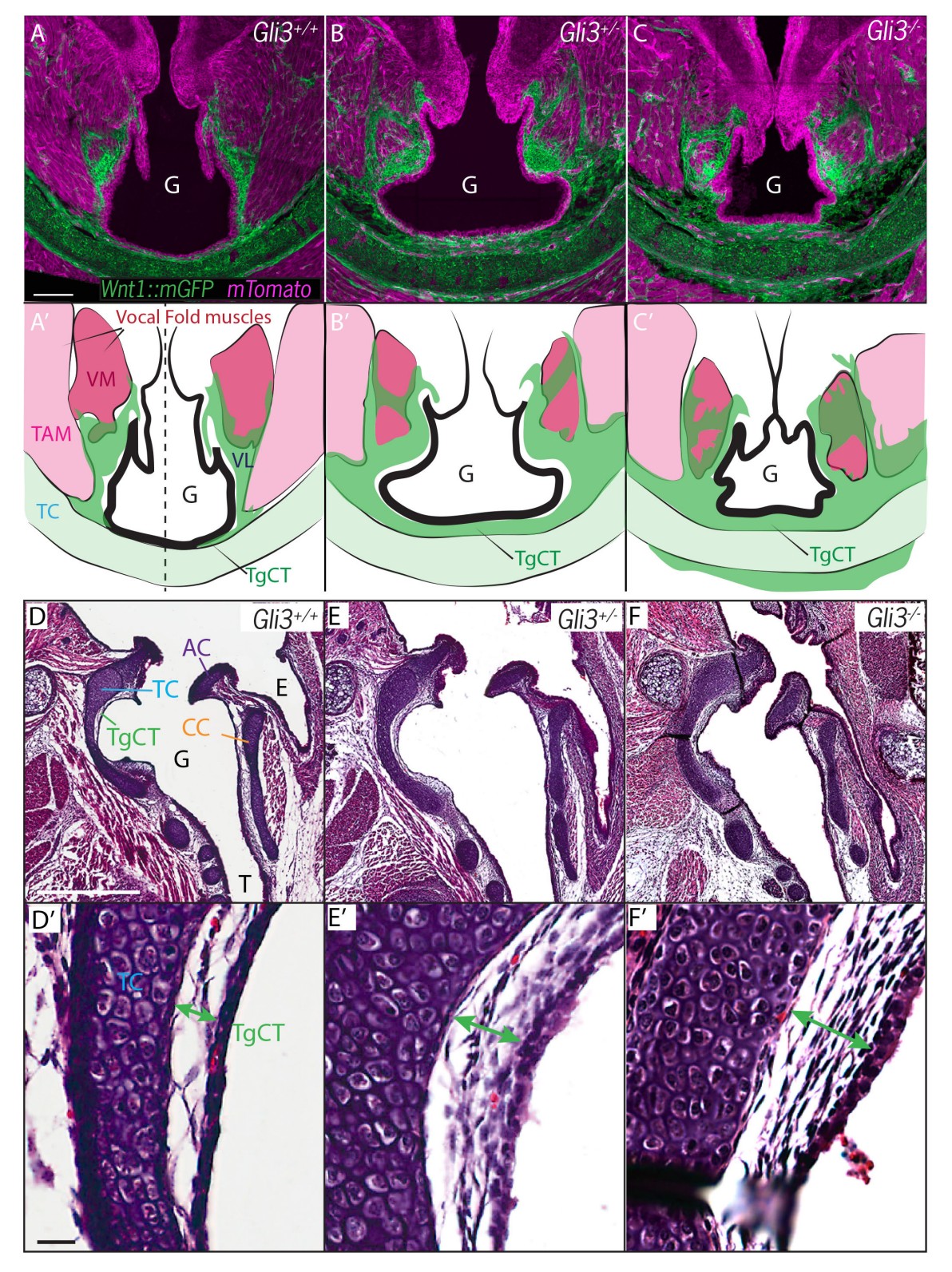

**Figure 6.** Expanded Thyroglottal connective tissue in *Gli3* mutants is neural crest derived. (**A–C**) Horizontal sections of the E18.5 ventral larynx in *Gli3+/+;Wnt1Cre/+; R26mTmG*(A) *Gli3+/-;Wnt1Cre/+; R26mTmG* (B) *Gli3-/-;Wnt1Cre/+; R26mTmG*(C) embryos. (**A'–C'**) Diagrams representing anatomy observed in (**A–C**). Black dotted line indicates sectional plane for (**D–F'**). (**D–F**) H&E staining of midline sagittal sections of E18.5 larynges in *Gli3+/+*(D), *Gli3+/-* (E); *Gli3-/-*
*Figure 6 continued on next page*

**Figure 6 continued**

(F) embryos. (D′–E′) Magnified view of Thyroglottal Connective tissue (green arrows) of the vocal pouch from sections in (D–F). Abbreviations: (E) Esophagus, (G) Glottis, (VL) Vocal Ligament, (VM) Vocalis Muscle, (TAM) Thyroarytenoid Muscle. Scale bars indicate 100 μm.

while the glottis remained absent in $Fuz^{-/-}Fgf8^{+/LacZ}$ mice, overall anatomy was improved, as cricoid and arytenoid cartilage elements could be identified (*Figure 2—figure supplement 1D,D′*). We also previously found that the high arched palate phenotype of *Fuz* mutant mice results from effects prior to neural crest specification, because mice with specific deletion of *Fuz* using a conditional allele driven in neural crest by $Wnt1^{cre}$ do not display high arched palate (*Tabler et al., 2013*). Likewise, we find here that laryngeal morphogenesis is largely normal in $Fuz^{flox/-};Wnt^{Cre/+}$ mice (*Figure 2—figure supplement 2*). Together, . . . . . .

## $Gli3^{xt-J}$ mutant mice display dose-dependent defects in laryngeal morphogenesis

The morphology of the larynx and vocal folds is complex, so in order to assess even subtle phenotypes in our mutant mice, we adopted a strategy of laryngeal morphometrics previously applied to human larynges (*Eckel and Sittel, 1995*). First, we quantified the morphology of the vocal folds themselves by measuring the cross sectional area occupied by the larynx, vocal muscles, and the $Wnt1^{Cre}$ labeled vocal ligaments (*Figure 7A,B,C*). Consistent with the observed excess neural crest discussed above, we detected a significant increase in the area occupied by the vocal ligament (*Figure 7C*). We observed no corresponding increase in the vocal muscle area, resulting in a significant change in the ratio of the area occupied by vocal ligament to that occupied by vocal muscles (*Figure 7D*). Strikingly, these phenotypes were dose dependent, with heterozygotes being significantly different from both wild-type and homozygotes (*Figure 7C,D*).

Next, we quantified the expansion of the TgCT by measuring the maximum distance between the dorsal edge of the thyroid cartilage and the ventral epithelial lining of the glottis (*Figure 7A,E*). Again, we observed a dose-dependent increase in this metric from wild-type to $Gli3^{xt-J}$ heterozygote to $Gli3^{xt-J}$ homozygotes (*Figure 7E*). The increase in connective tissue was observed along the length of the AP axis of the larynx in mutants, as evident in H&E stained sagittal sections of the larynx (*Figure 6D′–F′*).

Finally, we measured the area of the glottic opening, which was significantly reduced in homozygous animals compared to controls, but was not changed in heterozygous animals (*Figure 7F*). Together, these data demonstrate that heterozygous $Gli3^{xt-j}$ mutant mice display a milder version of the same laryngeal phenotype observed in the homozygotes.

## The acoustic structure of ultrasonic volcaizations is altered in heterozygous $Gli3^{xt/+}$ mutant mice

Ultimately, vocalizations are the functional output from the larynx and vocal folds, which manifests as audible speech in humans and as audible and ultrasonic cries in mice. Throughout life, mice use a variety of ultrasonic vocalizations (USV), from pup isolation calls to adult courtship displays (*Holy and Guo, 2005*; *Neunuebel et al., 2015*; *Noirot, 1966*; *Sewell, 1970*; *Zippelius and Schleidt, 1956*). Because $Gli3^{xt/+}$ heterozygous mice are viable and display mild defects in laryngeal morphology (above), we examined recordings of pup isolation calls for evidence of altered vocalization. We analyzed over 9000 vocalizations from 5 wild type (4718) and 6 heterozygous (4295) mouse pups (see *Figure 8A and B* for example spectrograms), finding no significant differences in vocalization duration ($F_{1,9}$ = 3.88, p=0.08) and mean frequency ($F_{1,9}$ = 0.12, p=0.73), but a significant difference in bandwidth ($F_{1,9}$ = 12.22, p=0.007) (*Figure 8C–E*).

Mouse USVs are known to be highly variable (*Heckman et al., 2016*), and it was unclear *a priori* what features of vocalizations might be modified, therefore in addition to measures of basic acoustic properties, we used an analysis method that takes into account the entire structure of the vocal repertoire to ask if subtler differences in vocal phenotype might be present in our mutant mice. From our 9000+ calls, we constructed a map of the vocal repertoire space in which vocalizations with similar frequency contours occupy adjacent regions in the map (*Figure 9A–C*; see Materials and

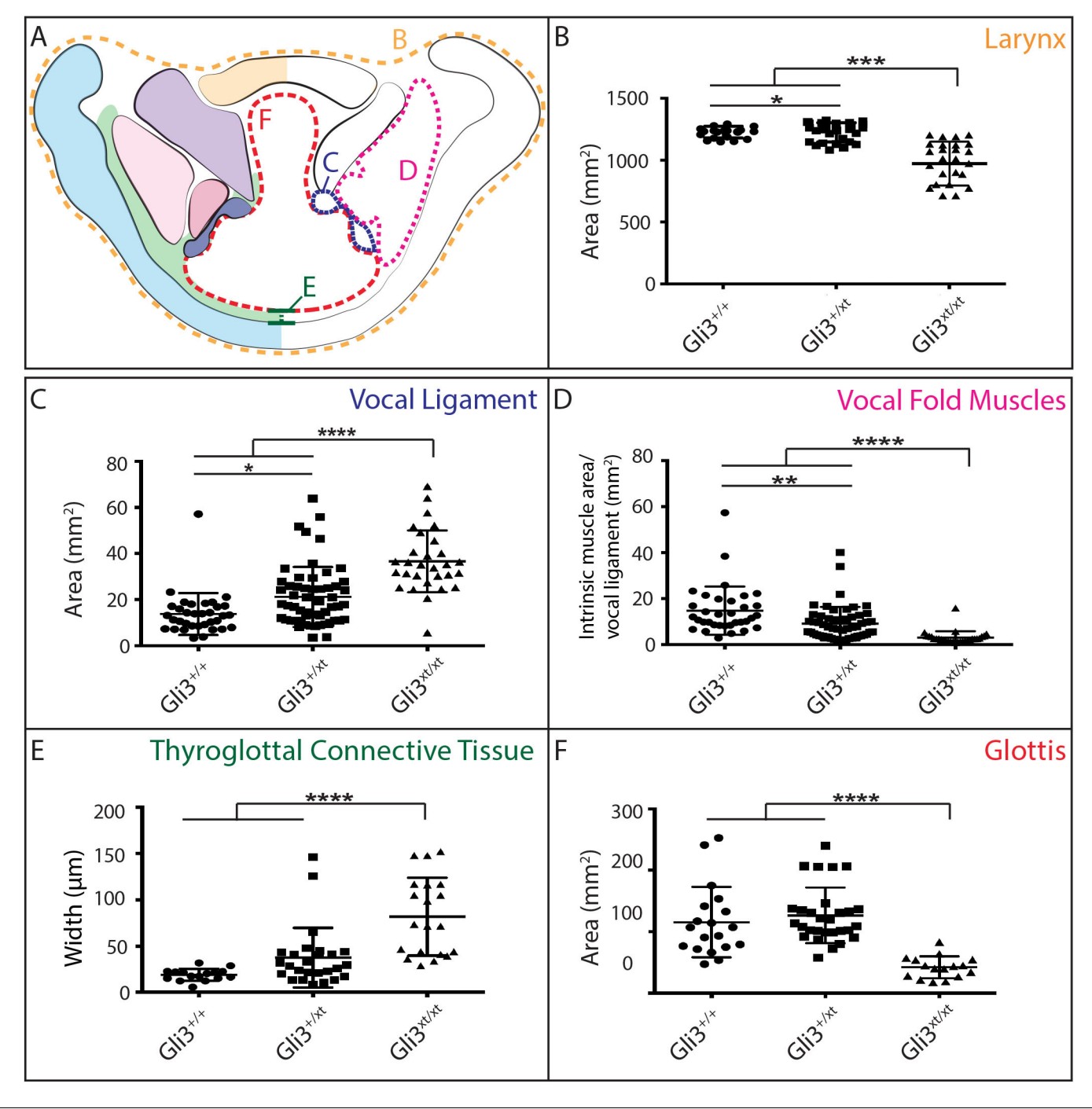

**Figure 7.** *Gli3* mutant laryngeal morphology is significantly altered. (**A**) Diagram representing laryngeal measurement presented in (**B–F**). (**B**) Quantification of total laryngeal cross sectional area excluding extrinsic muscles in E18.5 *Gli3$^{+/+}$* (*n* = 5), *Gli3$^{+/-}$* (*n* = 5), *Gli3$^{-/-}$* (*n* = 5) embryos (orange, (**A**). (**C**) Quantification of Vocal ligament area E18.5 *Gli3$^{+/+}$*, *Gli3$^{+/-}$*, *Gli3$^{-/-}$* embryos (blue, (**A**). (**D**) Quantification of vocal fold muscle area in E18.5 *Gli3$^{+/+}$*, *Gli3$^{+/-}$*, *Gli3$^{-/-}$* embryos. (**E**) Quantification of Thyroglottal connective tissue is in E18.5 *Gli3$^{+/+}$*, *Gli3$^{+/-}$*, *Gli3$^{-/-}$* embryos. (**F**) Quantification of Glottic space in E18.5 *Gli3$^{+/+}$*, *Gli3$^{+/-}$*, *Gli3$^{-/-}$* embryos. P values, * = 0.05, ** = 0.01, *** >0.001, **** >0.0001.

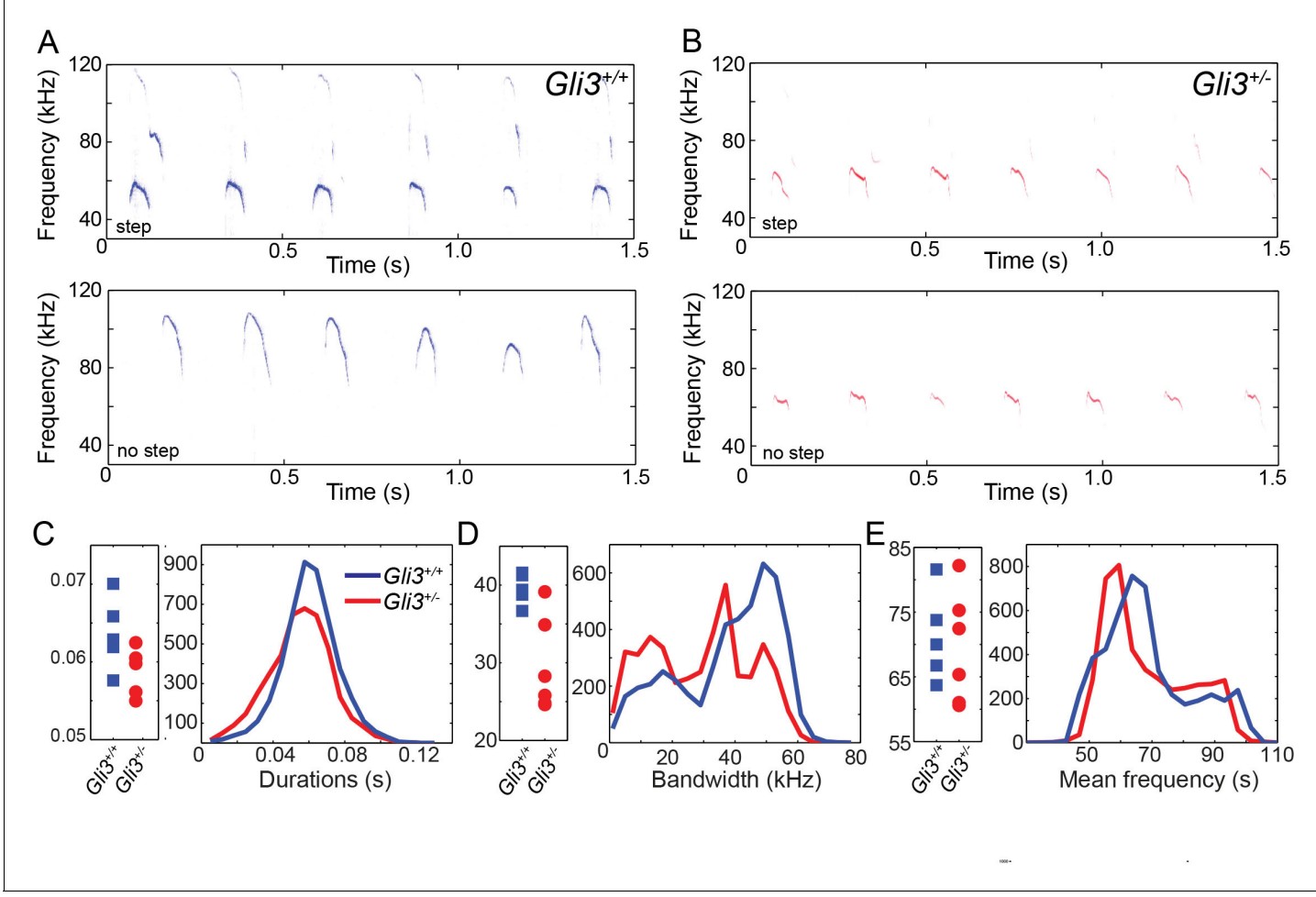

**Figure 8.** WT and HT vocalizations do not differ on simple acoustic measures. (**A**) Examples of *Gli3*[+/-] vocalizations with (top panel) and without (bottom panel) frequency steps. (**B**) Examples of *Gli3*[+/-] vocalizations with (top panel) and without (bottom panel) frequency steps. (**C**) Duration of *Gli3*[+/-] (blue) and *Gli3*[+/-] (red) pup vocalizations. Average values for each individual (left panel) and summary histogram of all vocalizations (right panel). (**D**) Bandwidth of *Gli3*[+/+] (blue) and *Gli3*[+/-] (red) pup vocalizations. Average values for each individual (left panel) and summary histogram of all vocalizations (right panel). (**E**) Mean frequency of *Gli3*[+/-] (blue) and *Gli3*[+/-] (red) pup vocalizations. Average values for each individual (left panel) and summary histogram of all vocalizations (right panel).

methods for details). In this map, simple calls lacking abrupt and discontinuous changes in frequency (frequency steps) cluster in the central body of the map, while more complex vocalizations with obvious frequency steps are distributed in 'islands' or 'peninsulas' surrounding the central body (*Figure 9C*).

This analysis revealed that the vocal repertoires of the two genotypes differed, and this difference was significant at multiple map locations (*Figure 9D–F*). While heterozygous *Gli3* mutant pups were capable of producing the full range of vocalizations made by control mice, they were significantly less likely to produce vocalizations with abrupt frequency steps (*Figure 9E,F*, warm colors), which is consistent with the observed reduction in overall vocal bandwidth in the mutant mice (*Figure 8D*). In our map, vocalizations in the same map area are acoustically similar, and so differences in the map between genotypes represent differences in acoustic structure between the genotypes; however, relative distance in the map beyond local relationships cannot be interpreted. For example, the vocalizations in the map area with n = 549 vocalizations differs from the vocalizations in the map area with n = 763 only by the presence or absence of a small initial high frequency component (present in n = 549, absent in n = 763). Whether this acoustic distinction is behaviorally relevant is as yet unknown.

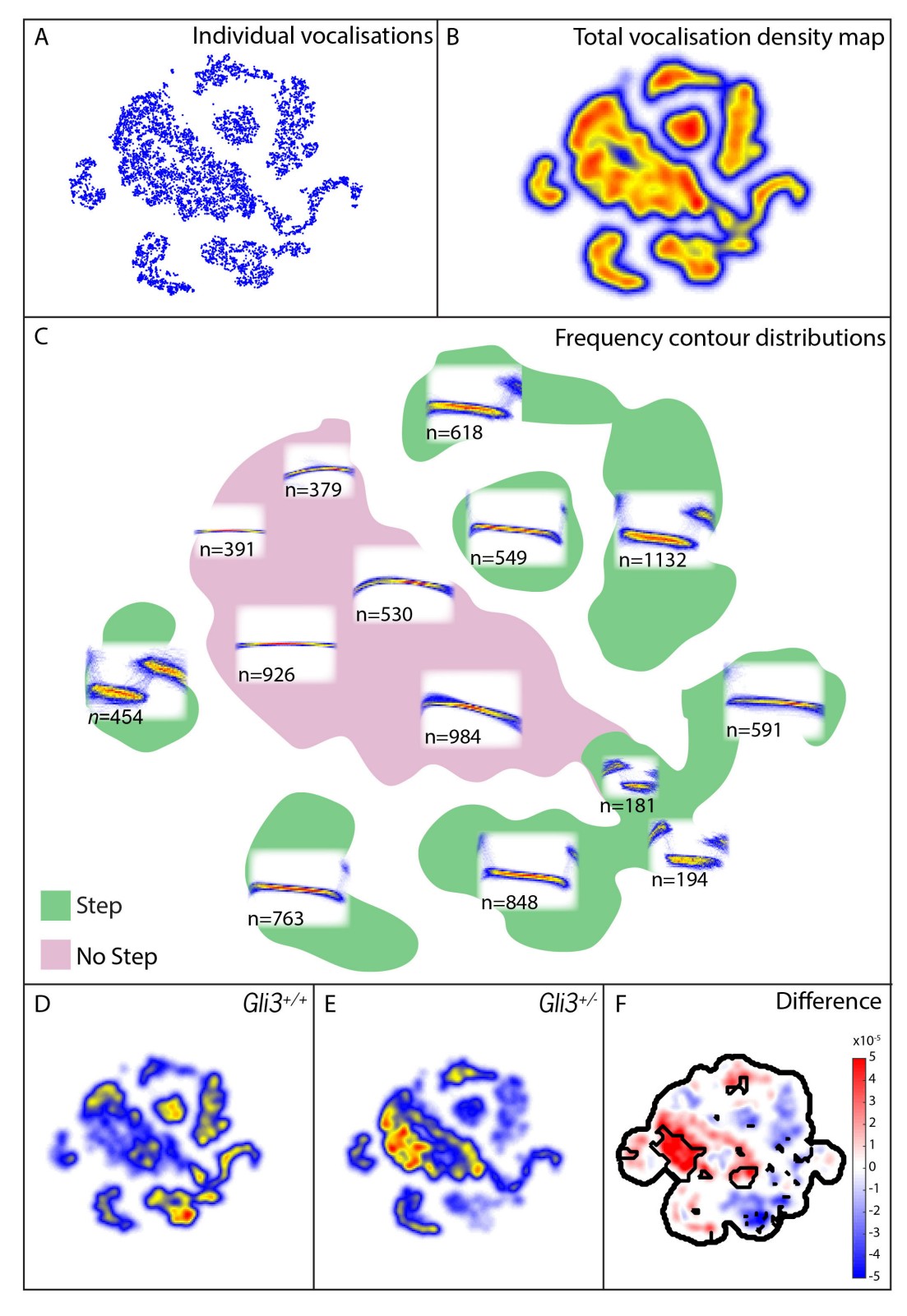

**Figure 9.** Map of pup vocal repertoire reveals differences in acoustic structure.  (**A**) Position of each vocalization within the vocal repertoire map (Individual vocalizations). To generate the map, we defined the difference between two vocalizations to be the dynamically time-warped (*Sakoe and Chiba, 1978*) mean squared error between them. Low-dimensional structure is then extracted from these distances using t-Distributed Stochastic Neighbor Embedding (t-SNE) [*van der Maaten and Hinton (2008)* and *Berman et al. (2014)*, resulting in the points seen in (**A**). This embedding

*Figure 9 continued on next page*

*Figure 9 continued*

results in a clustered structure, This two dimensional non-linear embedding preserves local neighbor relationships in the original high dimensional space. Because this embedding could be equivalently presented at any angle, the vertical and horizontal axes here are arbitrarily chosen and do not represent, for example, the leading directions of variation within the data set. (B) Estimated density of the vocalizations within the map (total vocalization density map). (C) Overlapped frequency contours for regions across the vocal repertoire map, showing the distribution of syllable types (Frequency Contour Distribution). (D) $Gli3^{+/+}$ vocalizations in the map. (E) $Gli3^{+/-}$ vocalizations in the map. (F) Difference between the maps with significance regions outlined. ($F_{1,9}$ = 0.04, p=0.85) were observed, and only non-significant changes we observed in frequency bandwidth ($F_{1,9}$ = 1.26, p=0.29) and average frequency ($F_{1,9}$ = 0.28, p=0.61).

To test more directly for differences in the number of step and non-step vocalizations between WT and $Gli3^{xt/+}$ each vocalization was labeled as a step or non-step based on their position in the vocal repertoire map (see *Figure 9C*, vocalizations in green regions were labeled step, vocalizations in the pink region were labeled non-step). We found that 69% of control vocalizations had steps (3257/4718) versus only 48% of mutant vocalizations (2073/4295). Conversely, mutant pups produced a far higher proportion of calls without steps (*Figure 9E*, warm colors). This difference in the proportion of step vocalizations was highly significant (chi2 = 400.42, df = 1, p<0.0001). In sum, $Gli3^{xt/+}$ heterozygous mice, display both morphological defects in the larynx and changes in their patterns of vocalization acoustics. While the acoustic structure of vocalizations is governed not only by the larynx, but also by the structure of the palate and pharynx, as well as neural inputs, our data nonetheless suggest that the mouse can provide a model for studying the links between laryngeal and voice defects in cilia and Gli-related craniofacial syndromes.

## Discussion

Despite the fundamental role of the larynx and vocal folds in mammalian communication, we still understand little about the molecular genetic control of their development. Here, we have explored the developmental basis for severe laryngeal defects in the ciliopathic *Fuz* mutant mice, and we describe qualitatively similar laryngeal defects, as well as vocalization defects, in mice mutant for the HH signal transduction protein Gli3. While we cannot rule out the possibility that *Fuz* loss may impact additional signaling activities, one parsimonious interpretation is that in the absence of *Fuz*, defective ciliogenesis leads in turn to defective Gli3 processing and disrupted HH signaling. Indeed, the key function of *Fuz* in both mice and frogs is the control of ciliogenesis (*Gray et al., 2009*; *Park et al., 2006*), and while Fuzzy controls PCP in *Drosophila* (*Collier and Gubb, 1997*), *Fuz* appears not to govern PCP-dependent processes in mice (*Heydeck and Liu, 2011*). Second, we have shown in *Fuz* mutant mice that both Gli3 processing and HH target gene expression are disrupted in the head during stages of neural crest cell specification and patterning (*Tabler et al., 2013*). Similar results have been reported for the limb and neural tube (*Gray et al., 2009*; *Heydeck et al., 2009*). Third, Gli3 is a known transducer of HH signals, and we recently showed that *Fuz* and *Gli3* mutant mice share a common phenotype of an enlarged crest-derived frontal bone in the skull (*Tabler et al., 2016*). That result in the skull parallels what we observed here in the larynx: A severe defect arising from the excess neural crest in *Fuz* mutants and a milder, but qualitatively similar, defect in the HH transducer *Gli3*. Combined with a central role for cilia in HH signal transduction (*Goetz and Anderson, 2010*), we feel the most parsimonious interpretation is that cilia-mediated HH signaling defects in *Fuz* mutants alter laryngeal morphogenesis.

Our data therefore support a model in which the expansion of laryngeal neural crest in *Fuz* mutants is a consequence of increased neural crest cell numbers at specification stages caused, ultimately, by expanded *Fgf8* expression. This conclusion is supported by our previous findings of increased neural crest cell numbers at E9 in Fuz mutant mice, as well as by our observation of excess *fgf8* gene expression in these mutants (*Tabler et al., 2013*) and by the finding in chick that Fgf8 can control neural crest cell numbers (*Creuzet et al., 2004*). Moreover, because *Fuz* is required for Gli3 processing (*Heydeck et al., 2009*; *Tabler et al., 2013*), our data are consistent with previous observations of expanded *fgf8* gene expression in *gli3* mutant mice (*Aoto et al., 2002*). Together, the data presented here and in our previous studies (*Tabler et al., 2013*, *2016*) suggest that

expanded neural crest is a unifying embryological mechanism for the palate, skull and larynx defects in ciliopathic mouse models.

That said, it is important to note that we cannot presently rule out additional, later roles for HH signaling in laryngeal morphogenesis. On the contrary, specific ablation of *smoothened* in post-specification neural crest using *Wnt1^Cre^* elicits a complete loss of the neural crest-derived thyroid cartilage (*Jeong et al., 2004*), though the effect of that manipulation on the vocal folds was not reported. Likewise, our deletion of *fuz* with *Wnt1^Cre^* also caused disruption of thyroid cartilage morphology (*Figure 2—figure supplement 2*), though this effect was less severe than that seen with smoothened disruption (*Jeong et al., 2004*).

Finally, *sonic hedgehog* is strongly expressed in the epithelial cells lining the glottis and the HH transducer *gli1* is expressed in the adjacent vocal folds in E11.5 mice (*Lungova et al., 2015*), suggesting another potential site of action. Future studies using conditional genetic approaches to assess the temporal and spatial requirements will be required before a comprehensive picture of HH signaling in the larynx can emerge.

Together, the data here provide new insights for future studies in two distinct areas. First, they shed light on the mechanisms of mammalian laryngeal development and suggest that the mouse larynx provides a tractable animal model for exploring the nexus between laryngeal structure and function. Second, they offer new insights into the role of neural crest in laryngeal morphogenesis, which may inform our understanding of the evolution of vocalization in vertebrates.

## Embryonic origins of the mammalian larynx and vocal folds

Fate mapping is a critical prerequisite for understanding the etiology of defects in development, so our understanding of laryngeal development has been hindered by a lack of directed fate mapping of this organ. In fact, our knowledge of lineage relationships in the mouse larynx comes only from tangential findings in studies focused on other topics. The fate mapping data presented here therefore provides substantial insight and serves as a useful complement to the thorough fate maps of pharyngeal regions in birds and amphibians. We consider several notable findings:

First, our data demonstrate a mixed origin for laryngeal cartilages. Indeed, we find a mixed lineage even within a single cartilage element, with the thyroid cartilage being predominantly, but not completely, derived from *Wnt1-cre*-labeled neural crest descendants (*Figure 3*). Specifically, the caudal ventral midline of this cartilage was of mesodermal origin, labeled by *Mesp1-cre* (*Figure 4*). This result adds the thyroid cartilage to the roster of individual skeletal elements assembled by fusion of crest and mesoderm-derived mesenchymal precursors (e.g. *Le Lièvre, 1978*; *Noden, 1988*). In addition, we found no evidence for a neural crest contribution to the arytenoid or cricoid cartilages, and instead our *Mesp1-cre* lineage data suggest a mesodermal origin for these elements (*Figures 3* and *4*).

These findings contradict a previous report suggesting a neural crest origin for all three laryngeal cartilages (thyroid, cricoid and arythenoid) with the anterior mesoderm boundary at the tracheal level (*Matsuoka et al., 2005*). However, several independent lines of evidence support our conclusion of distinct lineages for the laryngeal cartilages. First, we found no evidence for a neural crest contribution to these cartilages using either *Wnt1* or *Pax3* promoters for lineage labeling. Second, conditional deletion of *Smoothened* or *Sox9* using *Wnt1-cre* results in specific loss of the thyroid cartilage but leaves the cricoid and arytenoids intact (*Jeong et al., 2004*; *Mori-Akiyama et al., 2003*). A third line of evidence comes from whole animal loss-of-function for *R-spondin2*, which functions as a modulator of Wnt signaling throughout development (*de Lau et al., 2014*). In both *R-spondin2* mutants and *Rspondin2/Lrp6* double mutants, both cricoid and arytenoid cartilages are absent, while the thyroid cartilage remains unaffected (*Bell et al., 2008*; *Yamada et al., 2009*). Because neural crest and mesoderm-derived craniofacial structures are known to respond differently to Wnt signaling (e.g. *Barrell et al., 2012*, *Li et al., 2013*, *Quarto et al., 2009*), these data suggest distinct embryological origins for these cartilages. Fourth, our fate mapping data are consistent with avian fate maps in which the arytenoid and cricoid cartilages have a mesodermal origin, demonstrated both by transplantation and clonal analysis after retroviral labeling (*Evans and Noden, 2006*; *Noden, 1986a*). Finally, the amphibian pharyngeal skeleton, while considered to retain a more ancestral form, is nonetheless derived from a combination of neural crest and mesoderm (*Sefton et al., 2015*).

A second interesting finding concerns the cranial mesodermal origin for the vocal fold muscles (*Figure 4*). This finding is in contrast to what has been previously described; lineage analysis suggested a somitic origin of laryngeal muscles in both birds (*Couly et al., 1992*; *Huang et al., 1997*; *Noden, 1983*, *1986b*) and amphibians (*Piekarski and Olsson, 2007*). Our combined *Pax3^Cre* and *Isl1^Cre* lineage data indicate that the mammalian vocal fold muscles are not of somitic origin but derived from cranial mesoderm, consistent with recent studies performed in mice (*Gopalakrishnan et al., 2015*; *Lescroart et al., 2015*).

Finally, the data argue for a neural crest origin for diverse connective tissues in the larynx, including the vocal ligaments and thyroglottal connective tissue (*Figure 3*). These findings are significant because the viscoelastic properties of such connective tissues play an important role in sound production in mammals (see below).

Thus, our findings extend previous work highlighting the intricate interrelationship between migratory neural crest- and mesoderm-derived muscles during craniofacial morphogenesis (*Noden and Trainor, 2005*). Indeed, interactions between cranial mesoderm and cranial neural crest cells are essential for the normal patterning of the complex musculature of the head (*Grenier et al., 2009*; *Heude et al., 2010*; *Rinon et al., 2007*), perhaps explaining the lack of differentiated muscle in the crest-infused larynx of *Fuz* mutant mice (*Figure 2*). As such, our new data from the mammalian larynx complement existing work in other regions of the vertebrate head and highlight the key role of neural crest in the evolution of craniofacial morphology in general and the vocal apparatus specifically.

## Excess neural crest as a common feature underlying craniofacial ciliopathies

Human ciliopathies commonly involve craniofacial defects as well as laryngeal and voice defects, including breathy voices in Bardet-Biedl Syndrome and a hoarse voice in Joubert and Oral-Facial-Digital Syndromes. Moreover, laryngeal stenosis or narrowing is observed in ciliopathies (*Hayes et al., 2008*; *Silengo et al., 1987*), consistent with glottic narrowing in *Gli3* mutants. Interestingly, Barnes Syndrome is a clinical entity that very closely overlaps the spectrum of defects in the known ciliopathy Jeune syndrome, but with the addition of severe laryngeal defects (*Barnes et al., 1969*; *Burn et al., 1986*). The genetic basis for Barnes syndrome is unknown, but it is possible that mutations in *Fuz* or its interacting CPLANE proteins may be involved (*Toriyama et al., 2016*).

In all cases, the embryological basis for human laryngeal defects remains only very poorly understood, but data here and elsewhere argue that excess neural crest may be a central causative agent. For example, the severely deranged *Fuz* larynx was found to be filled with *Wnt1-cre* labeled crest-derived mesenchymal cells (*Figure 5*), which we interpret as a more severe version of the defect observed in *Gli3* mutants. Likewise, the high arched palate that characterizes diverse ciliopathies is also present in *Fuz* mutant mice, where it is accompanied by an excess of the neural crest (*Tabler et al., 2013*). Moreover, we have also recently described a novel skull defect in *Fuz* mutant mice in which mesoderm derived parietal bones of the skull fail to form at the expense of expanded neural crest-derived frontal bones (*Tabler et al., 2016*). Importantly, we also find that a milder version of that phenotype is present in *gli3* mutant mice (*Tabler et al., 2016*). We conclude then that laryngeal, palatal, and skull defects arising from defective cilia-mediated Gli signaling share a common etiology rooted in the excessive neural crest.

## Laryngeal developmental biology, neural crest, and the evolution of animal vocalization

Our work here focuses on a tractable model organism with well-developed genetic tools to explore the developmental biology of the mammalian larynx, revealing a key role for neural crest. An important implication of the work, however, is that similar studies in non-model animals could substantially advance our understanding of animal communication.

For example, the Panamanian tungara frog is a deeply studied model for evolution by sexual selection (*Ryan, 1985*). The complex mating call of the tungara frog is generated by a remarkable, sexually dimorphic elaboration of larynx called the fibrous mass (*Griddi-Papp et al., 2006*). Strikingly, the embryonic origins of the fibrous mass and the molecular genetic systems underlying its development are entirely unknown. However, understanding its morphogenesis is important, as the

final size and shape of the fibrous mass differs between related species in the *Physaleamus* genus, as do the calls produced by these species (*Ryan and Drewes, 1990*).

Reptiles provide another interesting context for future study. While alligators have a vocal folds relatively similar to that of mammals (*Riede et al., 2015*), snakes and tortoises have highly derived larynges, in which novel vibrating structures take the place of vocal folds. In bull snakes, defensive hissing sounds are generated by a flexible horizontal shelf in the larynx (*Young et al., 1995*), while in tortoises, sound appears to be generated by elastic bands on the lateral walls of the larynx (*Sacchi et al., 2004*). The embryonic tissue origins of these structures, as well as the molecular controls that guide their development, will be of interest.

In all three cases above, these laryngeal specializations are not muscular, but rather resemble connective tissue. Our finding of a neural crest origin for connective tissue in the mouse larynx suggests that novel vocalization structures in other animals may be crest derived. In light of the importance of neural crest in the diversification of vertebrate craniofacial structures (*Frisdal and Trainor, 2014*; *Le Douarin and Dupin, 2012*), we propose that a broader study of laryngeal developmental biology will shed light on the evolutionary diversification of vertebrate vocalization mechanisms.

## Form and function in the mammalian larynx

The ultrasonic calls of rodents have emerged as a useful model for studies of mammalian vocalization (*Arriaga et al., 2012*; *Fischer and Hammerschmidt, 2011*) (*Heckman et al., 2016*; *Portfors and Perkel, 2014*). Unlike the audible vocalizations generated by vibrations due to the pressure differential across the apposed vocal folds, rodent USVs are generated by a planar impinging air jet (*Mahrt et al., 2016*). Nonetheless, USVs are generated by the larynx and vocal fold adduction is an important factor both for sound production generally and for frequency modulation (*Johnson et al., 2010*; *Riede, 2013*). Indeed, direct imaging during USV production revealed a tight apposition of the vocal folds but an absence of vibrations normally observed during audible vocalization (*Sanders et al., 2001*). Moreover, many physiological parameters of mouse USV production parallel those of audible vocalization in other mammals (*Riede, 2011*, *2013*). Because so little is known about the etiology of human laryngeal birth defects and their relationship to voice dysfunction, we suggest that studies in mouse models will be informative.

We focused on pup isolation vocalizations, which are acoustically distinct from -and simpler than-adult ultrasonic vocalizations (*Liu et al., 2003*). Pup calls are also processed preferentially in mothers (*Elyada and Mizrahi, 2015*; *Liu and Schreiner, 2007*) and elicit maternal approach, retrieval and care (*Sales and Pye, 1974*). We found that $Gli3^{xt/+}$ pups produce vocalizations with durations and average frequencies that are not significantly different from those of their control littermates, however the bandwidth of these vocalizations is significantly different. Moreover, our more fine-grained analysis of vocalization shapes revealed that control and mutant mice differed in the proportion of specific vocalization types produced. In particular, $Gli3^{xt/+}$ mutation decreased the propensity of mice to make step vocalizations with abrupt frequency discontinuities, also known as 'punctuated' (*Panksepp et al., 2007*) or 'jump' (*Hanson and Hurley, 2012*) syllables.

The mechanisms by which such step syllables are generated remain unclear, but we consider two possible explanations for this phenotype. First, vocalization requires exquisite neural control (*Arriaga et al., 2012*; *Van Daele and Cassell, 2009*), and HH signaling is known to control neural patterning (*Briscoe and Thérond, 2013*). It may be, then, that alterations in neural pattern in these mice result in imperfect neural control of the larynx. However, our data on the overall acoustic structure of cells in the mutant mice argue against this explanation. For example, call duration in rats, another rodent with similar ultrasonic vocalizations (*Sales and Pye, 1974*), is precisely correlated with EMG activity of laryngeal muscles, and the activity patterns of these muscles during mouse ultrasonic vocalization reflect patterns seen in other mammals during audible phonation (*Riede, 2011*, *2013*). However, we found that duration of vocalizations was the same between control and mutant mice (*Figure 8*). In addition, disruption of the vocal center of the adult mouse cortex leads to changes in the distribution of mean frequencies of vocalizations (*Arriaga et al., 2012*), a parameter that was not affected in our $Gli3^{xt/+}$ mice (*Figure 8*). Conversely, manipulation of the vocal center did not alter the distribution of syllables produced (*Arriaga et al., 2012*), while *Gli3* mutation did (*Figure 9*). Finally, step vocalizations like those affected in our mutant mice are not correlated with either thyroarytenoid muscle EMG activity or sub-glottal pressure (*Riede, 2011*,

*2013*); and in fact, such step vocalizations can be produced independently of muscle or neural activity in excised bat larynges (*Kobayasi et al., 2012*).

These data suggest that such steps may result from a passive biomechanical effect in the larynx itself, leading us to prefer the alternative explanation that defective vocalization in *Gli3$^{xt/+}$* mice results from defects in the larynx. By disrupting the normally tight connection of vocal fold muscles to the thyroid cartilage, we propose that the excess neural crest-derived connective tissue observed in the larynges of *Gli3$^{xt/+}$* heterozygous mice may disrupt the biomechanics of the vocal folds and thereby impair normal sound production. Ultimately, further studies will be required to better define the source of vocalization defects in *Gli3* mutant mice. Importantly, however, the data here demonstrate that mouse models can both inform our understanding of mammalian vocalization and could also provide insights into the etiology of human laryngeal and voice defects.

## Materials and methods

### Mouse lines
The following mouse lines were used: *Gli3$^{xt-j}$* (*Hui and Joyner, 1993*; *Johnson, 1967*); *Wnt1-cre*: *Tg: (Wnt1-cre)11Rth* (*Danielian et al., 1998* PMID: 9636087); *Mesp1$^{Cre}$* (*Saga et al., 1999*); *Isl1$^{Cre/+}$*(*Srinivas et al., 2001*) *Pax3$^{Cre/+}$* (*Engleka et al., 2005*) and reporter line *R26$^{tdTomato}$*(Ai9; *Madisen et al., 2010*), *R26$^{mT/mG}$*: GT(Rosa)26Sortm4(ACTB-tdTomato-EGFP)Luo (*Muzumdar et al., 2007*), Fuz mutants: *Fuz$^{gt(neo)}$* (*Gray et al., 2009*). Genotyping was performed as described in original publications. All animal work was performed in accordance with approved IACUC protocols at the University of Texas at Austin.

### Histology and morphometrics
All immunohistochemistry, skeletal and histological staining were performed according to standard protocols. All embryos were collected in cold PBS and fixed in 4% paraformaldehyde. All embryos were sectioned horizontally at 18 µm for cryosections and 4 µm for paraffin sections. R26RmT/mG cryosections were stained with DAPI (1:1000) and then coverslipped with Vectashield (Vector Labs). Primary antibodies used for immunohistochemistry on cryosections: anti-human Desmin (D33, Dako). Secondary antibodies used were Alexafluor 488 (Biotechnology Company, Austin, TX, USA) at 1:500. Hematoxylin and eosin staining and trichrome staining (HT25A, Sigma) were performed at the Dell Pediatric Research Institute Tissue Processing Core. Area and width of morphological features were determined in Fiji using the freehand selection and straight line tools, respectively. Two-four representative sections were measured from each biological replicate. All histological sections were imaged with a 20X lens on a Scanscope (Aperio, Leica) and processed via ImageScope (Aperio, Leica) and Adobe Photoshop.

### Recording and extracting mouse vocalizations
Male *Gli3$^{/xt/+}$* mice were mated with Swiss Webster females (Charles River Laboratories). 5 days post-natal pups were separated from the mother and isolated on bedding in a recording chamber. Mouse vocalizations were recorded with an Ultrasoundgate 416 hr (Avisoft) sound recording system with a CM16/CMPA microphone (Avisoft) at a 250 kHz sampling rate and 16 bit resolution using Avisoft-RECORDER software, with the microphone suspended 5 cm from the pup. The start and stop times of ultrasonic vocalizations were automatically detected and frequency contours extracted using Ax (*Seagraves et al., 2016*; https://github.com/JaneliaSciComp/Ax). Briefly, time overlapped segments were Fourier transformed using multiple discrete prolate spheroidal sequences as windowing functions, followed by an F-test to identify time–frequency points with intensity significantly above noise (p<0.01). Acoustic segmentation was verified, and, if necessary, corrected manually. Signals that had exceeded the amplitude limit of the recording system (had 'clipped') were excluded from analysis.

### Generating map of vocal repertoire space
In order to compare the vocalizations of the two genotypes we create a single high-dimensional space that fully captures all the acoustic structure in the frequency contours. We then visualize that high dimensional space using dimensionality reduction to create a two-dimensional map of the vocal

repertoire. In this way, we are able to look at the vocal similarity across the vocal repertoire in the same reference frame for both genotypes. Frequency contours were mean frequency subtracted, and then all pairs of frequency contours were compared using dynamic time warping (*Sakoe and Chiba, 1978*) to create an all-to-all distance matrix (($9013 \times 9013$)/2 comparisons). The data in this high dimensional distance matrix were then embedded into two dimensions using t-SNE (*van der Maaten and Hinton, 2008*) (transition entropy = 5, relative convergence of the cost function to 0.0001). t-Distributed Stochastic Neighbor embedding (t-SNE) is a nonlinear embedding method that aims to preserve the local structure within a data set. This is achieved through placing points into a low-dimensional space such that points that were nearby in a higher-dimensional representation remain nearby in the new representation. Specifically, this embedding is calculated through optimally matching local similarity measures obtained in both the high and low dimensional spaces. Unlike other non-linear embedding approaches, this technique preserves clusters within a data set, but will allow for larger length scale distortions in order to obtain the desired dimensionality reduction. This is precisely the opposite of PCA, multi-dimensional scaling, or Isomap (*Tenenbaum et al., 2000*), which aim to preserve the global structure at the expense of local distortions. Because t-SNE preserves local neighbor relationships from the full-dimensional space of the frequency contour, regions in the map can be thought of as rough categories of vocalizations based on acoustic similarity. However, only local relationships are preserved, long-length-scale relationships are distorted, which means that the axes of the map are inherently arbitrary. A detailed description of applying t-SNE to a behavioral data set can be found in (*Berman et al., 2014*).

## Acoustic analyses

Simple measures of acoustic structure (duration, bandwidth, and average frequency were calculated from the automatically extracted contours). Differences between WT and HT in these simple measures were tested using single factor ANOVAs on average values for each individual. To test for differences in the number of step and non-step vocalizations between WT and HT each vocalization was labeled as a step or non-step based on their position in the vocal repertoire map. This labeling was automatic and blind to genotype. Regions of significant difference between the HT and WT maps were determined using bootstrapping, where we estimated the variation in the measured probability density functions due having a finite number of vocalizations in the data set. This was achieved through separately resampling the 2-D embeddings of the vocalizations for each case (WT and HT) with replacement 10,000 times and convolving each of these resampled data sets with a Gaussian of width 4 to create distributions, $q_{HT}(\rho|x,y)$ and $q_{WT}(\rho|x,y)$ for each of the PDFs at every point in space ($q_{HT}(\rho|x,y) \equiv Prob(\rho_{HT}(x,y) = \rho)$ and $q_{WT}(\rho|x,y) \equiv Prob(\rho_{WT}(x,y) = \rho)$). These spatially-varying PDFs were obtained by fitting a Gaussian mixture model to the sampled PDFs (up to three peaks, chosen at each point by maximizing the Akaike Information Criterion). As we assume that the two populations are sampled independently, the probability that $\rho_{HT}(x,y)$ is greater than $\rho_{HT}(x,y)$, defined here as $P_{HT}(x,y)$, is thus given by numerically integrating

$$P_{HT}(x,y) = \int_0^\infty \int_0^{\rho\_HT} q_{WT}(\rho_{WT}|x,y)q_{HT}(\rho_{HT}|x,y)d\rho_{WT} \, d\rho_{HT}.$$ Regions of significant difference are those where $P_{HT}(x,y)<\alpha$ or $P_{HT}(x,y)>1-\alpha$. Here, we used $\alpha = 0.05$, but corrected for multiple comparisons using the Šidák correction. We conservatively assume the number of comparisons to be $2^{\wedge}H$, where $H$ is the entropy of the original 2D embedding of our data set ($H = -\int\int \rho(x,y)log_2 \rho(x,y)dx \, dy$, where $\rho(x,y) = \frac{1}{2}(\rho_{HT}(x,y) + \rho_{WT}(x,y))$). To test for differences in the number of step and non-step vocalizations between WT and HT each vocalization was labeled as a step or non-step based on their position in the vocal repertoire map (see *Figure 9C*, vocalizations in green regions were labeled step, vocalizations in the pink region were labeled non-step). Because the map was generated using vocalizations from both genotypes, this labeling was automatic and blind to genotype. We then compared the number of step and non-step vocalizations in the two genotypes using the $\chi^2$ test.

## Acknowledgements

We thank D. Parichy and T. Arlson for critical reading. This work was supported by an NRSA to JT from the NIDCR (F32DE023272); funding to ST from the Institut Pasteur, Association Française

contre le Myopathies, and Agence Nationale de la Recherche (Laboratoire d'Excellence Revive, Investissement d'Avenir; ANR-10-LABX-73); and NIH R01HD073151 to SAV and R01HD085901 to JBW SERE is supported by the Howard Hughes Medical Institute; JBW was once an early career scientist of the HHMI.

## Additional information

### Funding

| Funder | Grant reference number | Author |
|---|---|---|
| National Institute of Dental and Craniofacial Research | F32DE023272 | Jacqueline M Tabler |
| National Institutes of Health | R01HD073151 | Steven Vokes |
| Howard Hughes Medical Institute | | Roian Egnor<br>John B Wallingford |
| National Institutes of Health | R01HD085901 | John B Wallingford |

The funders had no role in study design, data collection and interpretation, or the decision to submit the work for publication.

### Author contributions

JMT, ST, Acquisition of data, Analysis and interpretation of data, Drafting or revising the article, Contributed unpublished essential data or reagents; MMR, Acquisition of data, Analysis and interpretation of data; GJB, Analysis and interpretation of data, Contributed unpublished essential data or reagents; SG, EH, HAA-lami, RDF, CC, KJL, Acquisition of data; BZY, Analysis and interpretation of data; SV, Drafting or revising the article; SERE, Analysis and interpretation of data, Drafting or revising the article; JBW, Conception and design, Analysis and interpretation of data, Drafting or revising the article

### Author ORCIDs

John B Wallingford, http://orcid.org/0000-0002-6280-8625

### Ethics

Animal experimentation: This work was performed by protocols approved by UT Austin (IACUC protocol # AUP-2015-00105) and by King's College London (Animal Use Protocol PPL 70/7441).

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
