## [Decision Letter]

Thank you for submitting your article "Cilia-mediated Hedgehog signaling controls form and function in the mammalian larynx" for consideration by *eLife*. Your article has been reviewed by three peer reviewers, one of whom is a member of our Board of Reviewing Editors, and the evaluation has been overseen by Fiona Watt as the Senior Editor. The reviewers have opted to remain anonymous.

The reviewers have discussed the reviews with one another and the Reviewing Editor has drafted this decision to help you prepare a revised submission.

Summary:

In this paper from the Wallingford lab, Tabler et al. investigate the poorly understood development of the mouse vocal apparatus. They analyze the embryological origin of the main tissues in the mouse larynx (e.g., neural crest and mesoderm) using genetic fate mapping techniques (which of course rely heavily on the specificity of the Cre lines utilized). They also study the etiology of the developmental defects in mouse mutants for two genes, *Fuz* and *GLi3*. Finally, they analyze the acoustic function of the vocal apparatus in mice with mild laryngeal defects (i.e., *Gli3*hets) and after some quite sophisticated studies, manage to uncover a phenotype. This is an interesting and well executed study about an understudied yet important organ, presenting high quality data.

Essential revisions:

1) One main issue to be resolved is to provide better evidence for the conclusion that HH signaling is the main pathway affected during the development of the vocal apparatus in *Fuz* mutants (which provides the basis for the title of the paper).

The reviewers agreed that the authors should analyze the *Ptch1^-/+^* mice to determine whether the vocal apparatus phenotype is similar to the one in *Fuz* mutants; this would add some additional evidence to their hypothesis.

2) Previous work by this group has shown that *Fuz* mutants have altered Gli processing and that a high arched palate is due to a local increase in neural crest cells in the maxillary prominence, which can be rescued by reducing *Fgf8* gene dosage. Is this same mechanism, increased *Fgf8* expression, producing the increased neural crest?

If not, are other signaling pathways altered in this region that may affect the local population of neural crest in the laryngeal region?

Other comments are included below to help you optimize the paper but they do not need to be addressed experimentally.

[Editors' note: further revisions were requested prior to acceptance, as described below.]

Thank you for resubmitting your work entitled "Cilia-mediated Hedgehog signaling controls form and function in the mammalian larynx" for further consideration at *eLife*. Your revised article has been favorably evaluated by Fiona Watt as the Senior editor, a Reviewing editor, and two reviewers.

The manuscript has been improved but there are some remaining issues that need to be addressed before acceptance, as outlined below:

Please respond to this additional question. Why do the larynxes of *Fuz* mutants look profoundly abnormal in Figure 2 and Figure 4, but more mildly so in Figure 2—figure supplement 1? Are these mice of the same background? Is this degree of variability observed in the *Fuz* mutants even on an isogenic background? Comment on the expressivity of the phenotype will be of use to the reader.

With the addition of the *Fgf8* partial "rescue" data, the Discussion needs to better reflect the model being proposed. Is the primary effect of *Fuz* and *Gli3* mutation the generation of neural crest via expanded *Fgf8* expression, or is there a direct effect of HH signaling on laryngeal morphogenesis. This can be made clearer.

---

## [Author Response]

*Essential revisions:*

*1) One main issue to be resolved is to provide better evidence for the conclusion that HH signaling is the main pathway affected during the development of the vocal apparatus in Fuz mutants (which provides the basis for the title of the paper).*

*The reviewers agreed that the authors should analyze the Ptch1-/+ mice to determine whether the vocal apparatus phenotype is similar to the one in Fuz mutants; this would add some additional evidence to their hypothesis.*

Unfortunately, we do not have the *Ptch1* mice in our colony, nor was our collaborator able to able to perform this experiment for us in a timely manner (as she has for the Fuz/FGF8 experiments discussed below). We have no choice now but to proceed without this experiment, and we hope the Reviewers will accept the following: We have adjusted our manuscript to make clear that while cannot rule out the possibility of other effectors, the strong preponderance of evidence (in this paper and in other published works) strongly argues for the role for Fuz and Gli3 in HH signaling. First, we have shown in both mice and frogs that the key function of Fuz is in ciliogenesis, organelles well-known to be essential for HH signaling in mice. Moreover, while Fuz controls PCP in *Drosophila*, Aimin Liu’s group has published data that strongly argue against any role for Fuz in mammalian PCP (Heydeck, Dev. Dyn. 2011).

Second, we have shown in Fuz mutant mice that expression of the commonly used HH reporter genes *Gli1* and *Ptch1* are decreased in RT-PCRs at neural crest cell specification stages (E9.0), and that processing of the Gli3R protein in western blots is also reduced (Tabler et al. Dev. Cell 2013). Moreover, we have shown by in situ hybridization that the *Ptch1* expression domain is reduced and converges at the craniofacial midline in *Fuz*mutants (Tabler et al. Dev. Cell 2013). Third, Gli3 is a known transducer of HH signals and in a paper published recently, we showed that Fuz and Gli3 mutant mice share a common phenotype of an enlarged crest-derived frontal bone in the skull (Tabler et al., Dev. Biol. 2016). That result in the skull parallels what we observed here in the larynx: A severe defect arising from excess neural crest in Fuz mutant and a milder, but qualitatively similar, defect in the HH transducer Gli3. Combined with the key role for cilia in HH signal transduction, we feel the most parsimonious interpretation is that cilia- mediated HH signaling defects in *Fuz*mutants alter laryngeal morphogenesis.

Our manuscript has been altered to make these points explicitly.

*2) Previous work by this group has shown that Fuz mutants have altered Gli processing and that a high arched palate is due to a local increase in neural crest cells in the maxillary prominence, which can be rescued by reducing Fgf8 gene dosage. Is this same mechanism, increased Fgf8 expression, producing the increased neural crest?*

*If not, are other signaling pathways altered in this region that may affect the local population of neural crest in the laryngeal region?*

With our collaborator, Dr. Karen Liu (King’s College), we were able to perform this experiment. In *Fuz* mutants where *Fgf8* is reduced we observed a partial rescue of anatomical organization of the laryngeal cartilages (See new Figure 2—figure supplement 1).

In our previous work, we also noted that deletion of Fuz using *Wnt1^CRE^* did not elicit the high arched palate and skull defects on complete Fuz mutants, arguing that Fuz function was required prior to neural crest specification. We have now performed this experiment in the context of the larynx, and as for the skull and palate, we find that *Wnt1^CRE^* deletion of Fuz does not disrupt laryngeal morphogenesis (New Figure 2—figure supplement 2).

In the revised manuscript, we discuss these new data and place them in context, pointing out that they are consistent with our previous findings that reduction of *Fgf8* rescues palatal defects (Tabler et al. 2013) as well as skull defects (Tabler et al., Dev Biol. 2016) in Fuz mutant mice.

[Editors' note: further revisions were requested prior to acceptance, as described below.]

*The manuscript has been improved but there are some remaining issues that need to be addressed before acceptance, as outlined below:*

*Please respond to this additional question. Why do the larynxes of Fuz mutants look profoundly abnormal in Figure 2 and Figure 4, but more mildly so in Figure 2—figure supplement 1? Are these mice of the same background? Is this degree of variability observed in the Fuz mutants even on an isogenic background? Comment on the expressivity of the phenotype will be of use to the reader.*

We regret that the previous manuscript was not sufficiently clear on this point.

In fact, the phenotype is consistently severe, but the images in Figure 2—figure supplement 1 are frontal sections, while those in Figure 2 and Figure 4 are transverse sections. We have now altered the manuscript text to make this clear and to point out the clear separation of esophagus and glottis in controls and the lack of an apparent glottis in mutants. We also changed Figure 2—figure supplement 1 to make this point more clear.

These changes are in the Results section of the manuscript, as well as in the new Figure 2—figure supplement 1.

*With the addition of the Fgf8 partial "rescue" data, the Discussion needs to better reflect the model being proposed. Is the primary effect of Fuz and Gli3 mutation the generation of neural crest via expanded Fgf8 expression, or is there a direct effect of HH signaling on laryngeal morphogenesis. This can be made clearer.*

We have added text to the Discussion to address this point. We propose that cilia-mediated HH signaling plays a crucial early role in the generation of neural crest cell numbers, and on the basis of various findings, we also suggest that it play additional later roles. This latter speculation will require temporally- and spatially-controlled manipulations of HH signaling that we feel would be beyond the scope of the current work.

The new verbiage can be found in the Discussion section of the manuscript.